# Exploiting Synthetic Lethality between Germline BRCA1 Haploinsufficiency and PARP Inhibition in JAK2V617F-Positive Myeloproliferative Neoplasms

**DOI:** 10.3390/ijms242417560

**Published:** 2023-12-16

**Authors:** Max Bermes, Maria Jimena Rodriguez, Marcelo Augusto Szymanski de Toledo, Sabrina Ernst, Gerhard Müller-Newen, Tim Henrik Brümmendorf, Nicolas Chatain, Steffen Koschmieder, Julian Baumeister

**Affiliations:** 1Department of Hematology, Oncology, Hemostaseology, and Stem Cell Transplantation, Faculty of Medicine, RWTH Aachen University, 52074 Aachen, Germany; max.bermes@rwth-aachen.de (M.B.); mrodriguez@ukaachen.de (M.J.R.); mszymanskide@ukaachen.de (M.A.S.d.T.); tbruemmendor@ukaachen.de (T.H.B.); nchatain@ukaachen.de (N.C.); jbaumeister@ukaachen.de (J.B.); 2Center for Integrated Oncology Aachen Bonn Cologne Düsseldorf (CIO ABCD), 52074 Aachen, Germany; 3Confocal Microscopy Facility, Interdisciplinary Center for Clinical Research IZKF, RWTH Aachen University, 52074 Aachen, Germany; sabernst@ukaachen.de; 4Department of Biochemistry, Faculty of Medicine, RWTH Aachen University, 52074 Aachen, Germany; gmueller-newen@ukaachen.de

**Keywords:** myeloproliferative neoplasms, MPN, BRCA1, haploinsufficiency, olaparib, interferon-alpha, IFNα, DNA repair, homologous recombination repair, synthetic lethality

## Abstract

Myeloproliferative neoplasms (MPN) are rare hematologic disorders characterized by clonal hematopoiesis. Familial clustering is observed in a subset of cases, with a notable proportion exhibiting heterozygous germline mutations in DNA double-strand break repair genes (e.g., *BRCA1*). We investigated the therapeutic potential of targeting *BRCA1* haploinsufficiency alongside the *JAK2*V617F driver mutation. We assessed the efficacy of combining the PARP inhibitor olaparib with interferon-alpha (IFNα) in CRISPR/Cas9-engineered *Brca1^+/−^ Jak2*V617F-positive 32D cells. Olaparib treatment induced a higher number of DNA double-strand breaks, as demonstrated by γH2AX analysis through Western blot (*p* = 0.024), flow cytometry (*p* = 0.013), and confocal microscopy (*p* = 0.071). RAD51 foci formation was impaired in *Brca1*^+/−^ cells compared to *Brca1*^+/+^ cells, indicating impaired homologous recombination repair due to *Brca1* haploinsufficiency. Importantly, olaparib enhanced apoptosis while diminishing cell proliferation and viability in *Brca1*^+/−^ cells compared to *Brca1*^+/+^ cells. These effects were further potentiated by IFNα. Olaparib induced interferon-stimulated genes and increased endogenous production of IFNα in *Brca1*^+/−^ cells. These responses were abrogated by STING inhibition. In conclusion, our findings suggest that the combination of olaparib and IFNα presents a promising therapeutic strategy for MPN patients by exploiting the synthetic lethality between germline *BRCA1* mutations and the *JAK2*V617F MPN driver mutation.

## 1. Introduction

Myeloproliferative neoplasms (MPN) are a rare type of blood cancer and a disorder of excess progenitor cell production induced by clonal hematopoietic stem cells (HSCs). Among the group of the *BCR-ABL1*-negative MPN, the three most frequent subtypes are essential thrombocythemia (ET), polycythemia vera (PV), and primary myelofibrosis (PMF). With the exception of a minority of triple-negative patients, these disease entities are typically induced by one of three classical driver mutations in the genes encoding the tyrosine kinase Janus kinase 2 (*JAK2*), the chaperone calreticulin (*CALR*), or the thrombopoietin receptor *MPL*, the most frequent being the *JAK2*V617F mutation [1]. Although MPN patients generally exhibit a more favorable prognosis compared to other hematologic malignancies, they face an increased risk of severe thrombotic and hemorrhagic events, leading to increased mortality [2].

Additionally, secondary myelofibrosis or transformation into acute myeloid leukemia (AML) can occur, significantly worsening the prognosis [3]. Hence, there is an urgent need for effective treatment strategies to enhance patients’ quality of life and improve complication-free survival. Currently, allogeneic HSC transplantation remains the only curative therapeutic option. However, due to its high morbidity and mortality, only few patients are eligible for this intervention [4].

MPN mostly occur sporadically, but in 7.6% of the cases, MPN are associated with a familial incidence [5]. In these familial MPN, affected family members may present with different MPN subtypes or driver mutations, implying a shared genetic predisposition for acquiring these driver mutations [6]. Utilizing whole-exome sequencing, we could demonstrate that four out of five families with familial MPN presented with germline mutations in genes involved in DNA double-strand break (DSB) repair-associated genes (i.e., *BRCA1*, *BRCA2*, *ATM*, and *CHEK2*), suggesting that these germline mutations might increase the risk of acquiring a somatic MPN driver mutation [7].

*BRCA1* or *BRCA2* (*BRCA1/2*)-mutated cancer cells are effectively targeted by poly ADP ribose polymerase (PARP) inhibitors, leveraging the principle of synthetic lethality [8,9]. This concept was first described in 1922 by Calvin Bridges, who observed that the simultaneous loss of two distinct genes was lethal for flies, while the loss of one alone was not [10]. In the context of *BRCA1/2*-mutated cancer cells, PARP plays a pivotal role in DNA single-strand break (SSB) repair. Inhibiting PARP impedes the repair of DNA SSBs, causing them to progress into more severe DSBs. In *BRCA1/2*-mutated cells, characterized by defective homologous recombination repair (HRR), PARP inhibition induces a substantial load of DSBs that surpass their repair capacity. This results in the accumulation of DNA damage, ultimately triggering apoptosis through the phenomenon of synthetic lethality existing between *BRCA1/2* mutations and PARP inhibition.

This concept is already exploited clinically in familial breast and ovarian cancer, where patients often harbor a heterozygous *BRCA1/2* germline mutation and develop cancer after a somatic loss of the second allele. These cancer cells can be targeted specifically with PARP inhibitors, while non-malignant cells, which have retained one functional allele, remain unaffected [11]. While it is widely accepted that the loss of the second allele is a prerequisite for carcinogenicity in breast cancer and ovarian cancer, *BRCA1* haploinsufficient cells show functional deficits under challenging conditions, such as replicative stress [12,13]. In the context of MPN, driver mutations, such as *JAK2*V617F, induce replicative stress and genetic instability [14,15,16,17]. Building upon this knowledge, we postulated that PARP inhibition induces synthetic lethality, specifically in *BRCA1* haploinsufficient *JAK2*V617F-positive cells.

Recent research utilizing an ataxia telangiectasia mutated (*ATM*) model, with ATM serving as the apex kinase regulating BRCA1, has unveiled that DNA damage induced by *ATM* loss-of-function mutations primes the type I interferon system via the cyclic GMP-AMP synthase-stimulator of interferon genes (cGAS-STING) pathway [18]. Interferon-alpha (IFNα) is a widely used therapeutic option in the treatment of MPN and is recognized for its ability to selectively target malignant stem cells, evoke long-lasting deep molecular remissions, and induce DNA stress [19,20,21]. With this understanding, we hypothesized that MPN cells harboring heterozygous *BRCA1* mutations exhibit heightened responsiveness to IFNα treatment and that IFNα could potentially enhance the sensitivity of MPN cells with DSB repair gene haploinsufficiency to PARP inhibitors.

## 2. Results

### 2.1. DNA DSBs Are Elevated in Brca1^+/−^ Jak2V617F Cells and Further Amplified by Olaparib

The first aim of this study was to assess whether 32D *Jak2*V617F *Brca1*^+/−^ cells exhibited elevated numbers of DNA DSBs compared to their *Jak2*V617F *Brca1*^+/+^ counterparts. We also investigated whether olaparib, IFNα, or their combination induced more DSBs selectively in *Brca1*^+/−^ cells. For this purpose, cells were treated with olaparib and/or IFNα for 24 h and analyzed for the frequency of γH2AX-positive cells by flow cytometry (Figure 1A). While olaparib alone exhibited a stronger impact on inducing DSBs than IFNα, the combined treatment showed the most pronounced impact. When compared to *Brca1*^+/+^ control cells, percentages of *Jak2*V617F *Brca1*^+/−^ cells positive for γH2AX were significantly higher when treated with olaparib and the combination of olaparib and IFNα.

To confirm and complement the findings obtained from the flow cytometry analysis, we conducted Western blot experiments using *Jak2*V617F *Brca1*^+/+^ and *Brca1*^+/−^ cells after in vitro treatment with olaparib and/or IFNα for 4 h (Figure 1B–D). In *Brca1*^+/−^ cells, we observed a significant increase in γH2AX levels following treatment with DMSO, olaparib and the olaparib/IFNα combination, in comparison to *Brca1*^+/+^ cells (Figure 1B,C). Additionally, basal levels of protein poly ADP-ribosylation (PARylation), a post-translational modification at DNA lesions catalyzed by PARP, were significantly higher in the *Brca1*^+/−^ cells compared to the *Brca1^+/+^* cells (Figure 1B,D). As expected, PARylation was effectively inhibited by olaparib treatment.

### 2.2. Impaired HRR Mechanism and Suppressed Proliferation and Viability in Brca1^+/−^ Jak2V617F Cells upon Treatment with Olaparib and IFNα

To investigate the impact of *Brca1* haploinsufficiency on HRR within *Jak2*V617F cells, we analyzed the formation of γH2AX and RAD51 foci using immunofluorescence and confocal microscopy, following 24 h treatment with olaparib or DMSO. RAD51 foci are crucial indicators of HRR functionality and are commonly examined to assess impairment of this repair pathway [22]. As expected, olaparib treatment prominently triggered the formation of γH2AX foci (Figure 2A). Although not reaching statistical significance, there was a numerical enhancement in the induction of γH2AX foci in *Brca1*^+/−^ cells in comparison to *Brca1*^+/+^ cells (Figure 2B). When analyzing RAD51 foci numbers, we observed an induction by olaparib, which was significantly higher in *Brca1*^+/+^ cells (Figure 2C). Olaparib-treated *Brca1*^+/−^ cells displayed an increased number of γH2AX foci without a corresponding RAD51 focus compared with *Brca1*^+/−^ cells (*p* = 0.1378; Appendix A).

Having demonstrated the induction of DSBs by both olaparib and IFNα in *Jak2*V617F cells, with a notably higher effect in *Brca1*^+/−^ cells compared to *Brca1*^+/+^ cells, we examined the effects of olaparib and IFNα on cell proliferation and cell viability. Our findings revealed that olaparib, IFNα, and their combination significantly reduced cell viability and cell proliferation in *Brca1*^+/−^ *Jak2*V617F cells compared to *Brca1*^+/+^
*Jak2*V617F cells (Figure 3A and Appendix A). Most importantly, and in line with the findings of our preceding experiments, proliferation, and viability were considerably more impaired in *Brca1^+/−^* cells than in *Brca1^+/+^* cells when subjected to olaparib, IFNα, or the combination.

Thereafter, we studied the effect of olaparib and IFNα on *Jak2*V617F *Brca1*^+/−^ and *Brca1*^+/+^ cells using MTT assays as an indicator of cell viability, proliferation, and cytotoxicity (Figure 3B–D and Appendix A). These assays revealed that the reduction in metabolic activity was significantly more pronounced in *Jak2*V617F *Brca1*^+/−^ cells than in *Jak2*V617F *Brca1*^+/+^ cells when treated with olaparib (Figure 3B and Appendix A), IFNα (Figure 3C and Appendix A), or the combination of both drugs (Figure 3D). We also analyzed the effect of olaparib and IFNα on the metabolic activity of 32D *Jak2* wildtype (WT) cells. Interestingly, olaparib and IFNα exhibited a weaker impact on the metabolic activity of *Jak2*WT cells, with olaparib still demonstrating a stronger effect on *Brca1*^+/−^ cells than on *Brca1*^+/+^ cells (Appendix A). However, for IFNα, no difference between *Brca1*^+/−^ and *Brca1*^+/+^ cells was observed. Moreover, by comparing the mean relative absorbances from the MTT assays with *Jak2*WT cells to those with *Jak2*V617F cells (Appendix A), we found that both olaparib and IFNα led to a more pronounced impairment of metabolic activity in *Jak2*V617F than in *Jak2*WT cells. Moreover, olaparib treatment had a stronger impact on *Jak2*V617F cells compared to *Jak2*WT cells, which was not observed for IFNα treatment.

To investigate the potential of olaparib and IFNα to induce apoptosis in *Jak2*V617F *Brca1*^+/−^ and *Brca1*^+/+^ cells, we utilized Annexin V/7-AAD apoptosis assays (Figure 3E). IFNα induced apoptosis to a greater extent than olaparib, whereas the combination provoked the strongest response. By calculating the coefficient of drug interaction (CDI), we observed higher synergistic effects of olaparib and IFNα in *Brca1*^+/−^ cells than in *Brca1*^+/+^ cells in MTT (Appendix A) and apoptosis (Appendix A) assays.

To assess the impact of olaparib and IFNα on the cell cycle of *Jak*2V617F *Brca1*^+/+^ and *Brca1*^+/−^ cells, we analyzed the fraction of cells in G_0_/G_1_ or G_2_/M phase with FxCycle Violet by flow cytometry after 24 h of treatment with olaparib, IFNα, or their combination. A pronounced cell cycle arrest in G_0_/G_1_ was induced in *Brca1*^+/−^ cells following single treatments of olaparib and IFNα, and this effect was notably enhanced by the combination treatment (Appendix A). In contrast, the reduction in the fraction of *Brca1*^+/+^ cells in G_2_/M phase was less pronounced. Additionally, we analyzed the expression of negative cell cycle regulators, specifically p16 (i.e., *CDKN2A* gene) and p21 (i.e., *CDKN1A* gene). p16, also known as cyclin-dependent kinase inhibitor 2A, inhibits the progression from the G_1_ phase to the S phase, and p21, also called cyclin-dependent kinase inhibitor 1, is involved in inhibiting the progression through the G_1_, the S, and the G_2_ phases. The upregulation of p16 and p21 was significantly more pronounced in *Brca1*^+/−^ cells when treated with both olaparib and IFNα, corroborating the flow cytometric cell cycle analysis (Appendix A).

In summary, the results affirm our hypothesis that *Jak2*V617F *Brca1*^+/−^ cells exhibit a higher number of DSBs compared with *Brca1*^+/+^ cells. Notably, we found that this susceptibility to DSBs is specifically heightened in *Jak2*V617F *Brca1*^+/−^ cells through PARP inhibition. Furthermore, olaparib and IFNα preferentially compromised the metabolic activity, proliferation, viability, and cell cycle progression of *Jak2*V617F *Brca1^+/−^* cells. These findings underscore the elevated sensitivity of *Brca1*^+/−^ cells to these therapeutic interventions.

### 2.3. Olaparib Induces IFNα Signaling via Activation of the cGAS-STING Pathway Specifically in Brca1^+/−^ Jak2V617F Cells

Intriguingly, recent evidence has suggested a link between DNA damage and the activation of the cGAS-STING pathway, which is one major pathway responsible for driving the production of IFNα in response to cytosolic microbial and self-DNA [23]. This pathway is part of the innate immune system and detects cytosolic microbial DNA but can also be activated by an accumulation of DNA damage, leading to the release of DNA into the cytoplasm and subsequent activation of the cGAS-STING pathway resulting in increased production of IFNα [18]. Therefore, we postulated that (a) the cGAS-STING pathway is constitutively active in *Jak2*V617F *Brca1^+/−^* cells and further augmented by olaparib treatment and (b) that the activated STING-pathway results in an elevated production of IFNα (Figure 4).

To test the hypothesis that the STING pathway is activated by cytoplasmic DNA as a result of increased DSBs in *Jak2*V617F *Brca1*^+/−^ cells, we analyzed transcriptional levels of *Sting1* and several interferon-responsive genes (*Stat1*, *Irf7*, *Mx1*, *Oas1a,* and *Isg15)* in *Jak2*V617F *Brca1*^+/−^ and *Jak2*V617F *Brca1*^+/+^ cells after treatment with olaparib, IFNα, or the combination (Figure 5). As expected, we observed an upregulation of all interferon-responsive genes after treatment with IFNα, and mRNA levels of *Sting1* were not elevated in *Brca1*^+/−^ cells upon treatments but even downregulated in the basal condition. However, interestingly, olaparib treatment induced the expression of *Irf7*, *Mx1*, *Oas1a,* and *Isg15* in *Jak2*V617F *Brca1*^+/−^ cells compared to *Jak2*V617F *Brca1*^+/+^ cells. Moreover, the basal expression of *Isg15* and *Mx1* was elevated in *Brca1^+/−^* compared to *Brca1^+/+^* cells, and when treated with both olaparib and IFNα, *Stat1*, *Oas1a,* and *Isg15* exhibited significant upregulation in *Brca1*^+/−^ cells.

To confirm that the induction of the interferon-responsive genes was indeed a result of cGAS-STING pathway activation, we conducted a secretion assay, as this pathway is also known to induce heightened expression and secretion of IFNα. Supernatant harvested from olaparib- or DMSO-treated *Jak2*V617F *Brca1*^+/−^ and *Brca1*^+/+^ cells, was added to freshly seeded *Jak2*V617F *Brca1*^+/+^ cells. The expression of interferon-responsive genes was then assessed by RT-qPCR to determine whether olaparib-treated cells had secreted IFNα (Figure 6A). We found that, indeed, basal levels of *Oas1a* and *Isg15* were upregulated in those cells that have received supernatant from *Jak2*V617F *Brca1*^+/−^ cells compared to *Jak2*V617F *Brca1*^+/+^ cells. This effect was further enhanced by olaparib treatment. Importantly, this effect was not observed when the cells were preincubated with an IFNαR1 blocking antibody before adding the supernatant, confirming that *Brca1* haploinsufficiency in *Jak2*V617F *Brca1*^+/−^ cells stimulates an increased production of IFNα.

Finally, to verify that the upregulation of interferon-responsive genes and the increased production of IFNα are indeed mediated through the cGAS-STING pathway, we treated *Jak2V617F Brca1*^+/−^ and *Brca1*^+/+^ cells with olaparib and the STING inhibitor H-151. Subsequently, we analyzed the expression of three interferon-stimulated genes (ISGs): *Mx1*, *Oas1a,* and *Isg15* (Figure 6B). Intriguingly, we observed that the increased ISG expression levels in *Brca1^+/−^* cells after treatment with olaparib were antagonized by STING inhibition, except for *Oas1a*. These findings indicate that DNA damage induced by PARP inhibition leads to an activation of the cGAS-STING pathway with subsequent increased production of IFNα in *Jak2*V617F-positive *Brca1* haploinsufficient cells. This unique interplay potentially renders these cells more susceptible to IFNα and other MPN-directed treatments.

## 3. Discussion

Loss-of-function mutations of DNA repair-associated genes play a role in many types of cancer, but their potential significance in MPN has only recently been suggested by a whole-exome sequencing study of our group, in which heterozygous germline mutations in DSB repair genes have been identified in four out of five families with familial MPN [7]. Moreover, even in patients with sporadic MPN, the natural incidence of germline mutations in DSB repair genes would be expected to be at least 0.1–0.5% since this is the combined incidence of *BRCA1* and *BRCA2* mutations in the general population [24]. As it is already known that cancers harboring germline mutations in DSB repair-associated genes are susceptible to PARP inhibition, it is tempting to consider whether this principle could also be translated to patients with familial or sporadic MPN. Currently, no clear recommendations on how to manage these MPN patients are available.

In this study, we demonstrated that heterozygous *Brca1* mutations in 32D *Jak2*V617F cells result in impaired HRR, suggesting a potential therapeutic vulnerability that could be exploited by PARP inhibitors. As previously reported, JAK2V617F induces DNA damage [14,15], and our findings suggest that *Jak2*V617F-positive cells harboring an additional heterozygous *Brca1* mutation experience even greater DNA damage, surpassing the capacity of the defective DSB repair machinery. This effect is notably amplified by olaparib, with significantly elevated γH2AX levels as demonstrated by Western blot and flow cytometry, further supported by close-to-significant immunofluorescence analyses (*p* = 0.07). Based on these findings, we propose that PARP inhibition could offer a promising therapeutic strategy for familial MPN patients carrying DSB repair-associated germline mutations by exploiting synthetic lethality. Furthermore, we observed an activation of the STING pathway in olaparib-treated *Brca1*^+/−^ *Jak2*V617F cells, leading to an increased secretion of IFNα. This cytokine, which has emerged as a standard treatment in MPN, exhibited enhanced efficacy in *Brca1*^+/−^ *Jak2*V617F cells compared to *Brca1*^+/+^ cells. While IFNα is capable of inducing long-lasting deep molecular remission and promoting the cycling of dormant stem cells [25,26], the combination of IFNα with olaparib presents a potential synergistic drug combination worthy of further investigation.

IFNα selectively induces cycling of the *JAK2*V617F HSC, leading to cell cycle stress-associated genomic instability and increased susceptibility towards PARP inhibition, particularly in cells with defective DNA repair mechanisms. To our knowledge, this therapeutic approach of combining PARP inhibitors with IFNα has not been investigated yet and holds potential for application in non-familial MPN patients as well, as *BRCA1* is epigenetically inactivated in 40% of all MPN samples analyzed [22], and alterations in DNA repair genes are a frequent feature in MPN patients [27]. This suggests that effective treatment options targeting defective DNA repair mechanisms might extend beyond familial MPN cases to encompass a larger cohort of MPN patients who could benefit from this targeted therapeutic approach. Supporting this notion, the effectiveness of PARP inhibitors in MPN with no detected DSB repair-associated gene mutation has already been demonstrated [28].

The cGAS-STING pathway has been recognized as an activator of the antitumor immune response [29], and in triple-negative breast cancer, the efficacy of olaparib depends on the activation of the cGAS-STING pathway, which recruits CD8^+^ T cells into the tumor microenvironment, thereby triggering an antitumor immune response [30]. Our findings indicate that olaparib induces an upregulation of the cGAS-STING pathway in *Brca1*^+/−^ cells, leading to an increase in intrinsic IFNα production.

Our study on murine *Jak2*V617F-positive 32D cells indicates the potential relevance of *Brca1* haploinsufficiency to human MPN disease. Given the rarity of MPN and the low prevalence of *BRCA1* mutations in the general population, identifying MPN patients harboring *BRCA1* mutations is challenging. To address this, we are actively conducting an extensive screening on patients with familial MPN. Subsequently, we plan to validate our mechanistic findings in further studies, utilizing primary MPN patient samples harboring *BRCA1* mutations and haploinsufficient *Brca1* animal models to assess how the concept of synthetic lethality between BRCA1 haploinsufficiency and PARP inhibition translates clinically into JAK2V617F-driven MPN. Considering that *BRCA1* mutations in patients with familial MPN extend beyond the bone marrow, data from triple-negative breast and ovarian cancer patients harboring heterozygous germline *BRCA1* mutations who have undergone olaparib treatment might aid in estimating treatment-associated side effects on *BRCA1*-haploinsufficient non-cancerous cells. Nevertheless, our MTT assays have already demonstrated that both olaparib and the combination with IFNα exert a more pronounced effect on *Brca1^+/−^* cells with an additional *Jak2*V617F driver mutation when compared to *Brca1^+/−^ Jak2*WT cells. Although PARP inhibitors are already approved for the treatment of other cancers and their efficacy and safety have been assessed in clinical trials [31,32], in vivo studies are required to assess the efficacy and safety of PARP inhibitors, both alone and in combination with IFNα, for the treatment in MPN. Additionally, considering that PARP inhibitors may impair DNA damage repair and pose a carcinogenic risk for homologous recombination proficient cells [33], further studies are required to estimate the risk of developing secondary cancers. This is underlined by reports suggesting an increased risk of developing AML or myelodysplastic syndrome in breast or ovarian cancer patients treated with PARP inhibitors over extended periods [34]. In conclusion, our findings suggest the potential of combining olaparib and IFNα as a promising therapeutic strategy in MPN patients by exploiting the synthetic lethality between germline *BRCA1* mutations and the *JAK2*V617F MPN driver mutation.

## 4. Materials and Methods

### 4.1. Cell Lines

32D *Jak2*V617F *Brca1*^+/−^ cells were generated by CRISPR/Cas9 using two guideRNAs targeting *Brca1* exon 10 (CD.Cas9.LFMV2350.AA (AGTCCAAAGGTGACAGCTAA) and CD.Cas9.LFMV2350.AB (GGTTAAGCGCGTGTCTCAAG) and Cas9 nuclease according to the manufacturer’s protocol (IDT technologies, Coralville, IA, USA). Parental 32D cells (RRID:CVCL_0118, DSMZ, Braunschweig, Germany) were retrovirally transduced with pMSCV-*Jak2*V617F-IRES-GFP. Clones were screened by Sanger sequencing (genomic DNA and mRNA) for frameshift mutations inducing premature stop codons in the same region as the human *BRCA1* c.2722G>T Glu908* missense mutation that was identified in our whole-exome sequencing analysis in familial MPN [7]. Off-target analysis was performed as described in the Appendix A and relevant off-target effects of the CRISPR/Cas9 process were ruled out. In all experiments, two different clones of each genotype were examined.

### 4.2. Flow Cytometry Analysis of DSBs

We treated, 2 × 10^6^ cells in a concentration of 1 × 10^6^ cells/mL with either DMSO (Serva, Heidelberg, Germany) or olaparib (10 µM) (Selleckchem, Cologne, Germany) and with H_2_O or IFNα (10,000 U/mL) in RPMI-1640 (PAN-Biotech, Aidenbach, Germany) medium with 10% fetal calf serum (FCS; PAN-Biotech), 5 ng/mL murine interleukin 3 (mIL-3, ImmunoTools, Friesoythe, Germany), and 1% penicillin/streptomycin (Gibco-Thermo Fisher Scientific, Waltham, MA, USA) for 24 h. The positive control was treated with etoposide (10 µM) (Merck, Darmstadt, Germany) for 4 h, and the negative control was treated with the ATM inhibitor Ku55933 (10 µM) (Selleckchem) for 4 h. Cells were fixed and permeabilized using the FIX and PERM Cell Fixation and Cell Permeabilization Kit (Thermo Fisher Scientific, Waltham, WA, USA) and stained with Phospho-Histone H2A.X (γH2AX, Ser139) monoclonal antibody (CR55T33, PE) (eBioscience, Frankfurt am Main, Germany) and FxCycle Violet stain (Thermo Fisher Scientific). Cells were analyzed in a Gallios flow cytometer (Beckman Coulter, Pasadena, CA, USA) and analyzed with FlowJo (LLC; v10, BD Life Sciences, Franklin Lakes, NJ, USA). A detailed protocol is provided in the Appendix A.

### 4.3. MTT Assay

We performed 3-(4,5-dimethylthiazol-2-yl)-2,5-diphenyltetrazolium bromide (MTT) assays as previously published but with 1.5 × 10^4^ cells per well instead of 10^4^ cells in RPMI-1640 medium with 10% FCS, 5 ng/mL mIL-3 and 1% penicillin/streptomycin [35].

### 4.4. SDS-PAGE and Western Blot

We treated 2 × 10^6^ cells in a concentration of 1 × 10^6^ cells/mL for 4 h with IFNα (10,000 U/mL), olaparib (10 µM), or the combination in RPMI-1640 medium with 10% FCS, 5 ng/mL mIL-3, and 1% penicillin/streptomycin. Generation of lysates, SDS-PAGE, and Western blots were performed as previously published [36]. The used antibodies are listed in the Appendix A.

### 4.5. Apoptosis Assay

We treated 4 × 10^5^ cells in a concentration of 2 × 10^5^ cells/mL for 48 h with olaparib (10 µM) or IFNα (10,000 U/mL) in RPMI-1640 medium with 10% FCS, 5 ng/mL mIL-3, and 1% penicillin/streptomycin and then, 1 × 10^6^ cells were subjected to apoptosis assays, using the APC Annexin V Apoptosis Detection Kit (BioLegend, San Diego, CA, USA), with the samples being analyzed in triplicates with a Gallios flow cytometer (Beckman Coulter) and FlowJo (LLC; v10).

### 4.6. Cell Proliferation Assay

The different cell clones were seeded in a concentration of 2 × 10^5^ cells/mL in RPMI-1640 medium with 10% FCS, 5 ng/mL mIL-3, and 1% penicillin/streptomycin and treated with the following drugs: IFNα (10,000 U/mL), olaparib (10 µM), or the combination of IFNα and olaparib. Then, the cells were analyzed with a CASY-TTC cell analyzer (OMNI Life Science, Bremen, Germany) after 24 h, 48 h, and 72 h.

### 4.7. RT-qPCR

Reverse transcriptase quantitative PCR (RT-qPCR) for the analysis of cDNA transcripts was performed as previously published (20). Primer sequences are given in the Appendix A.

### 4.8. Confocal Microscopy Assay

We treated 1 × 10^5^ cells/mL with either olaparib (10 µM) or DMSO in RPMI-1640 medium with 10% FCS, 5 ng/mL mIL-3, and 1% penicillin/streptomycin for 24 h. Then, the cells were harvested, washed with PBS 1X, and centrifuged using Cytospin 4 centrifuge (Thermo Fisher Scientific) at 650 rpm for 5 min. After fixation with 4% paraformaldehyde for 20 min, the cells were permeabilized with 0.5% Triton X-100 for 10 min, blocked with 5% BSA for 1 h, and incubated with primary antibodies (γH2AX 1:500 Cell Signaling #9718, Danvers, MA, USA; RAD51 1:500 Invitrogen MA1-23271) at 4 °C overnight. Subsequently, the cells were washed and incubated with fluorochrome-conjugated secondary antibodies (anti-mouse Alexa Fluor 488 1:200 and anti-rabbit Alexa Fluor 594 1:200) for 1 h at room temperature, and nuclei were stained with Hoechst 33342 (Thermo Fisher Scientific). Images were captured with a Zeiss LSM 710 confocal microscope (Carl Zeiss GmbH, Jena, Germany) using ZEN black 2.3 SP1 software (version 14.0.27.201, Carl Zeiss GmbH). At least 200 cells in nine different areas of the sample were analyzed to quantify RAD51 and γH2AX foci. For γH2AX/RAD51 quantification, the percentage of cells with ten/five (respectively) or more foci per nucleus was scored.

### 4.9. Secretion Assay

For every clone, 5 × 10^6^ cells were cultured in 5 mL RPMI-1640 medium with 10% FCS, 1% penicillin/streptomycin, mIL-3 (5 ng/mL) and olaparib (10 µM) or DMSO for 24 h. Then, cells were washed twice with PBS and resuspended in 3 mL RPMI-1640 medium with 10% FCS, 1% penicillin/streptomycin and mIL-3 (5 ng/mL) and incubated for 4 h. Cells were centrifuged at 400× *g* and the supernatant was isolated, sterile-filtered, and one half of the supernatant was added to 5 × 10^5^ 32D *Jak2*WT *Brca1*^+/+^ cells preincubated for 30 min with an IFNα receptor blocking antibody (10 µg/mL). As a control group, the other half of the supernatant was added to 5 × 10^5^ 32D *Jak2*WT *Brca1*^+/+^ cells, and then, the cells were incubated for 24 h. Afterwards, RNA was isolated with the RNA purification kit from Macherey-Nagel (Düren, Germany) to generate cDNA and perform qPCRs.

### 4.10. Cell Culture

We acquired 32D cells from the DSMZ and cultured them in RPMI-1640 medium with 10% WEHI-3B supernatant as mIL-3 source, 10% fetal calf serum, and 1% penicillin/streptomycin. Cells were cultured at 37 °C with 5% CO_2_. For the assays, to have a steady mIL-3 concentration, the RPMI-1640 medium was supplemented with 5 ng/mL mIL-3 instead of 10% WEHI-3B supernatant.

### 4.11. Statistical Analysis

Statistical analysis was performed using GraphPad Prism 8.4.0 (GraphPad Software, Boston, MA, USA) to calculate independent *t*-tests or, in case of multiple comparisons and, unless otherwise stated, two-way ANOVAs (Tukey post hoc test). The results are displayed as individual values with mean and standard deviation (SD) if not indicated otherwise.

## 5. Conclusions

In summary, our study has demonstrated that *Brca1* haploinsufficiency induces DNA damage in *Jak2*V617F-positive cells while priming the type I interferon system via STING, rendering them more susceptible to olaparib treatment, whether applied as a standalone therapy or in conjunction with IFNα. This combined therapeutic approach, which targets DSB repair mechanisms, presents the potential for the treatment of MPN patients with germline DSB repair gene mutations, including those with familial and sporadic MPN. Further investigations are needed to fully elucidate the clinical implications of our findings and to assess the feasibility of implementing genetic counseling for DSB repair germline mutational testing in all MPN patients [7].

## Figures and Tables

**Figure 1 ijms-24-17560-f001:**
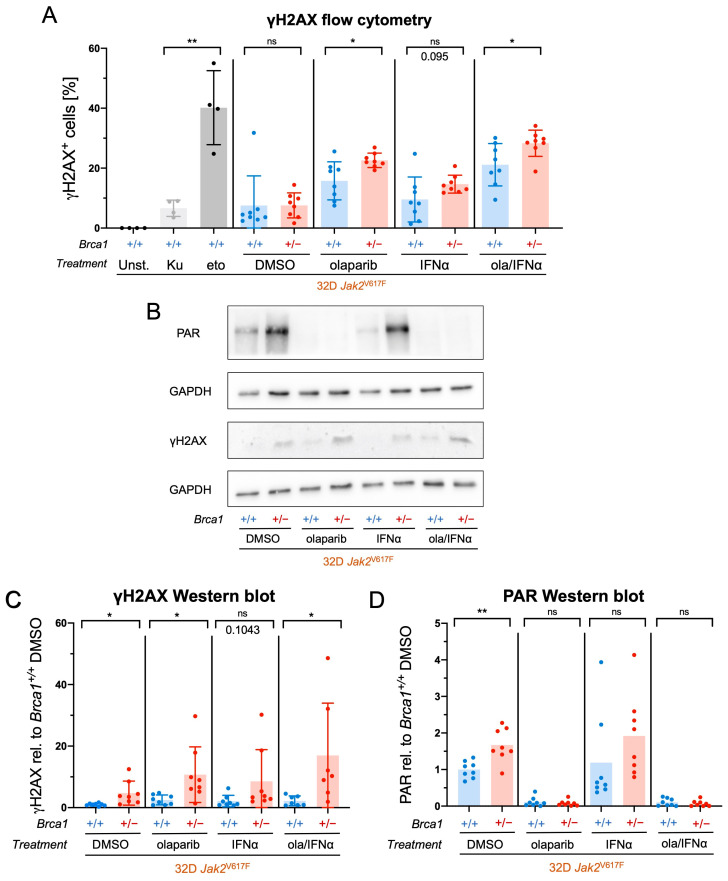
Enhanced double-strand break induction by olaparib in *Brca1*^+/−^ *Jak2*V617F cells. (**A**) *Jak2*V617F *Brca1*^+/+^ and *Brca1*^+/−^ cells were treated with olaparib (ola, 10 µM), interferon-alpha (IFNα, 10,000 U/mL), or DMSO for 24 h. As a negative control, cells were treated with the ATM inhibitor Ku55933 (Ku, 10 µM) and as a positive control, cells were treated with the topoisomerase II inhibitor etoposide (eto, 10 µM). After treatment, the cells were stained with a phospho-histone H2A.X (Ser139) monoclonal antibody and FxCycle™ Violet stain and then analyzed by flow cytometry (*n* = 4 of two clones respectively, exemplary flow cytometry plots see Appendix A). (**B**) Protein levels of γH2AX and protein PARylation were analyzed in *Jak2*V617F *Brca1*^+/+^ and *Brca1*^+/−^ cells after incubation for 4 h with olaparib (10 µM), IFNα (10,000 U/mL), and DMSO by Western blot (WB) using GAPDH as a loading control. (**C**) γH2AX and (**D**) PAR densitometric analyses from four independent WB experiments (relative to GAPDH as loading control and relative to *Brca1*^+/+^ DMSO, *n* = 4 with two clones each). Data are presented as mean ± SD and significances defined as: * *p* < 0.05 and ** *p* < 0.01, ns means no significance.

**Figure 2 ijms-24-17560-f002:**
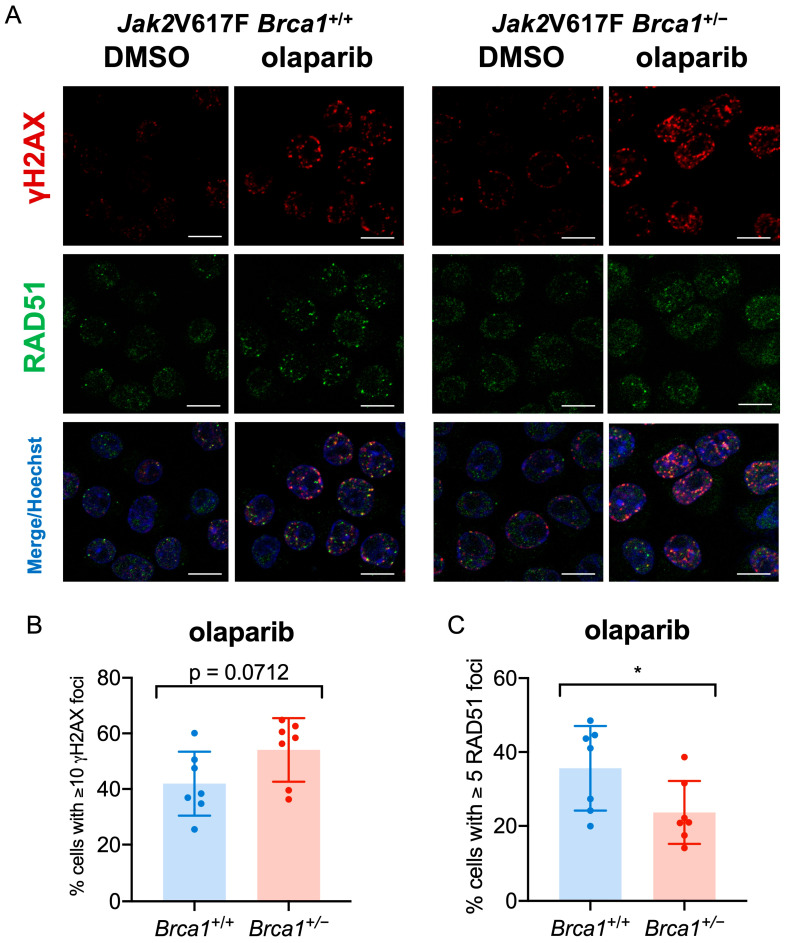
Impaired homologous recombination repair mechanism in olaparib-treated *Jak2*V617F *Brca1*^+/−^ cells. (**A**) *Jak2*V617F *Brca1*^+/+^ and *Brca1*^+/−^ cells have been treated for 24 h with DMSO or olaparib (10 µM), stained with anti-RAD51 and anti-γH2AX antibodies, incubated with anti-mouse Alexa Fluor 488 and anti-rabbit Alexa Fluor 594 and counterstained with Hoechst 33342. Images were taken with a Zeiss LSM 710 confocal microscope using Zen software (scale bar = 10 µm). (**B**) The percentage of cells with ≥10 γH2AX foci was scored of at least 200 cells per condition (*n* = 4). (**C**) The percentage of cells with ≥5 RAD51 foci was scored at least 200 cells per condition (*n* = 4). Data are presented as mean ± SD and significances defined as: * *p* < 0.05.

**Figure 3 ijms-24-17560-f003:**
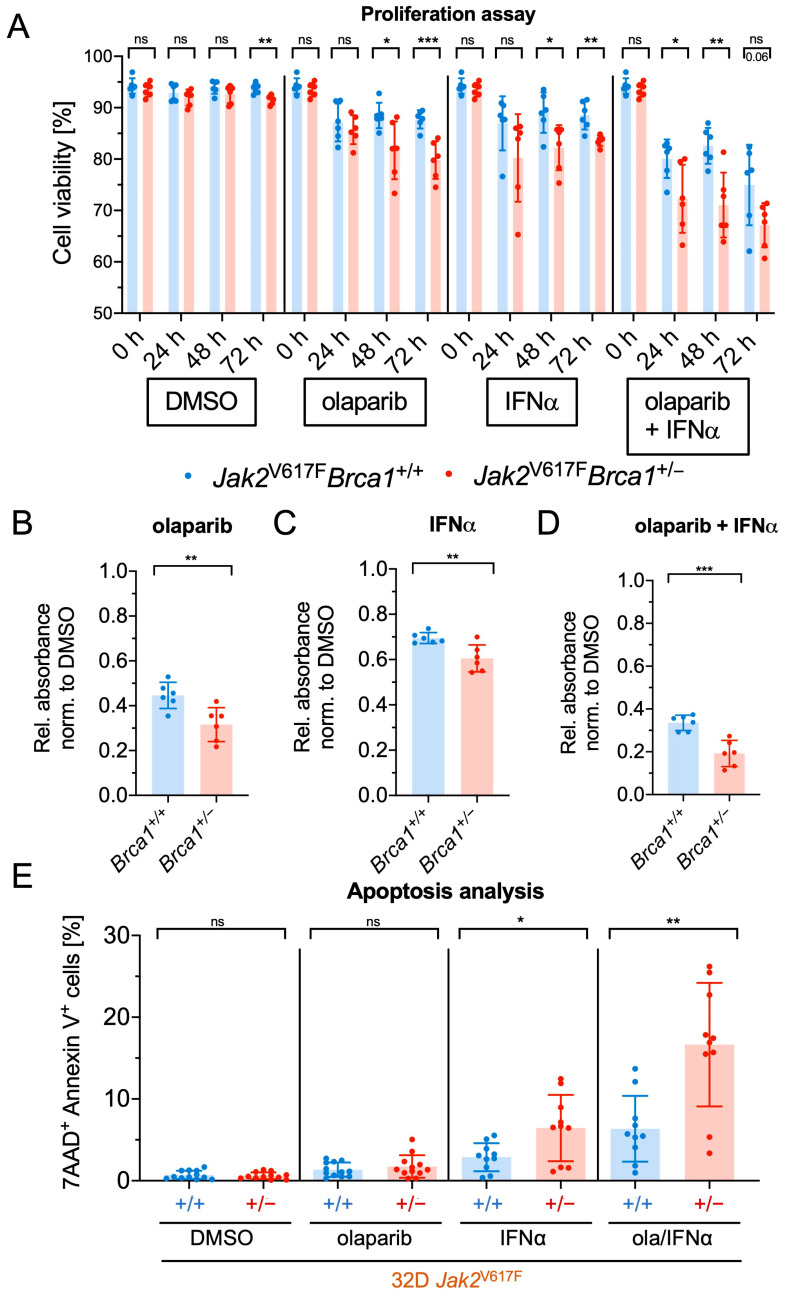
Olaparib and IFNα reduce cell proliferation and viability and induce apoptosis preferentially in *Jak2*V617F *Brca1*^+/−^ cells. (**A**) Cell viability was analyzed in *Jak2*V617F *Brca1*^+/+^ and *Brca1*^+/−^ cells treated with olaparib (ola, 10 µM), interferon-alpha (IFNα, 10,000 U/mL) or DMSO by analysis with a CASY cell counter after 24 h, 48 h, and 72 h (*n* = 3 with two clones each). (**B**–**D**) Metabolic activity was assessed by MTT assays in *Jak2*V617F *Brca1*^+/+^ and *Brca1*^+/−^ cells after treatment for 48 h with DMSO or increasing concentrations of IFNα and an additional fixed concentration of olaparib (10 µM) (*n* = 3 each). The relative absorption of *Brca1*^+/+^ and *Brca1*^+/−^ cells was compared after treatment with olaparib (10 µM) (**B**), with IFNα (10,000 U/mL) (**C**), and with the combination of olaparib (10 µM) and IFNα (10,000 U/mL) (**D**). (**E**) Apoptosis was analyzed in 32D *Jak2*V617F *Brca1*^+/+^ and *Brca1*^+/−^ cells treated with olaparib (10 µM), IFNα (10,000 U/mL) or DMSO for 48 h using AnnexinV-APC/7-AAD through flow cytometry. The experiment was done in triplicates (*n* = 5 with two clones each). Data are presented as mean ± SD and significances defined as: * *p* < 0.05, ** *p* < 0.01, and *** *p* < 0.001, ns means no significance.

**Figure 4 ijms-24-17560-f004:**
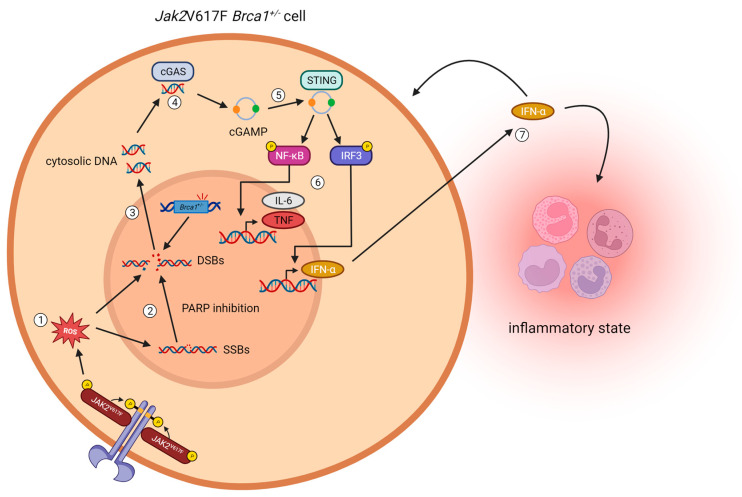
Hypothesis: PARP inhibition activates the cGAS-STING pathway in *Jak2*V617F *Brca1*^+/−^ cells. (1) The *Jak2*V617F mutation leads to the formation of reactive oxygen species and thereby causes DNA stress, resulting in an increased number of single- and double-strand breaks (SSBs/DSBs). (2) PARP inhibition suppresses the repair of SSBs, resulting in the formation of double-strand breaks (DSBs) that also cannot be repaired sufficiently due to the *Brca1* haploinsufficiency. (3) The accumulation of DSBs results in the translocation of DNA fragments into the cytoplasm. (4) cGAS recognizes cytosolic DNA and, upon detection of cytosolic DNA, synthesizes cGAMP. (5) cGAMP is detected by STING and, together with TBK1, phosphorylates and activates the transcription factors NF-κB and IRF3. (6) NF-κB and IRF3 translocate into the nucleus and induce the production of proinflammatory cytokines (e.g., TNF, IL-6, type I interferons). (7) Those cytokines are secreted into the interstitium where they bind to the membrane receptors of the original and the surrounding cells and hence induce a proinflammatory state. Created with Biorender.com (accessed on 17 October 2023).

**Figure 5 ijms-24-17560-f005:**
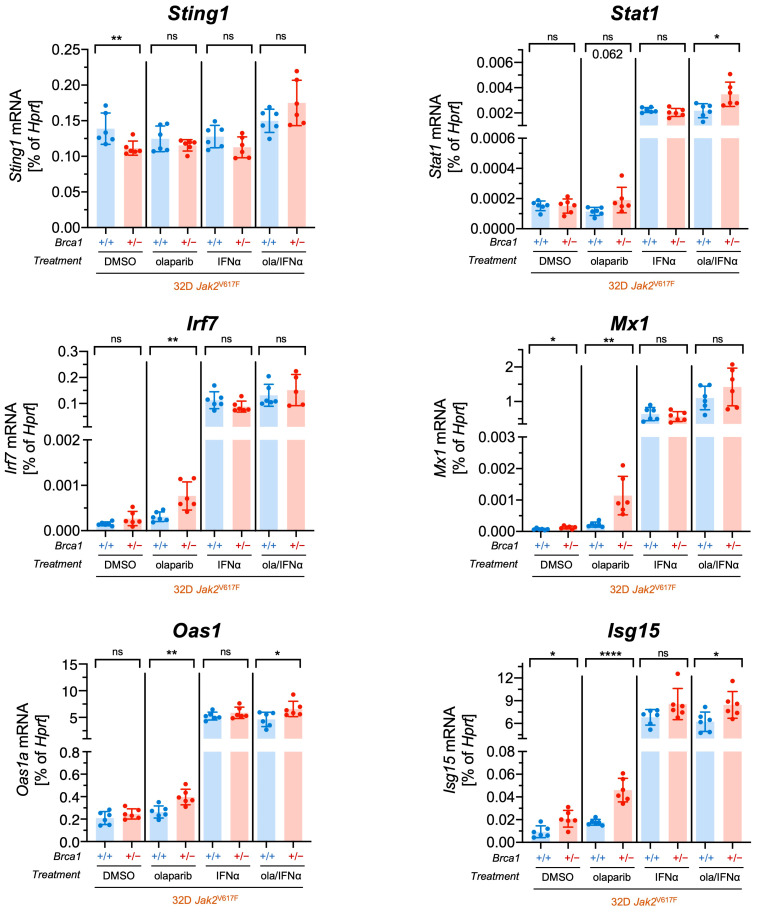
Induction of cGAS-STING pathway target genes by olaparib in *Brca1^+/−^ Jak2*V617F cells. mRNA levels of *Sting1*, *Stat1*, *Irf7*, *Mx1*, *Oas1*, and *Isg15* in *Jak2*V617F *Brca1*^+/+^ and *Brca1*^+/−^ cells treated for 24 h with olaparib (ola, 10 µM), interferon-alpha (IFNα, 10,000 U/mL), the combination, or DMSO (*n* = 3 with two clones each). Data are presented as mean ± SD and significances defined as: * *p* < 0.05, ** *p* < 0.01, and **** *p* < 0.0001, ns means no significance.

**Figure 6 ijms-24-17560-f006:**
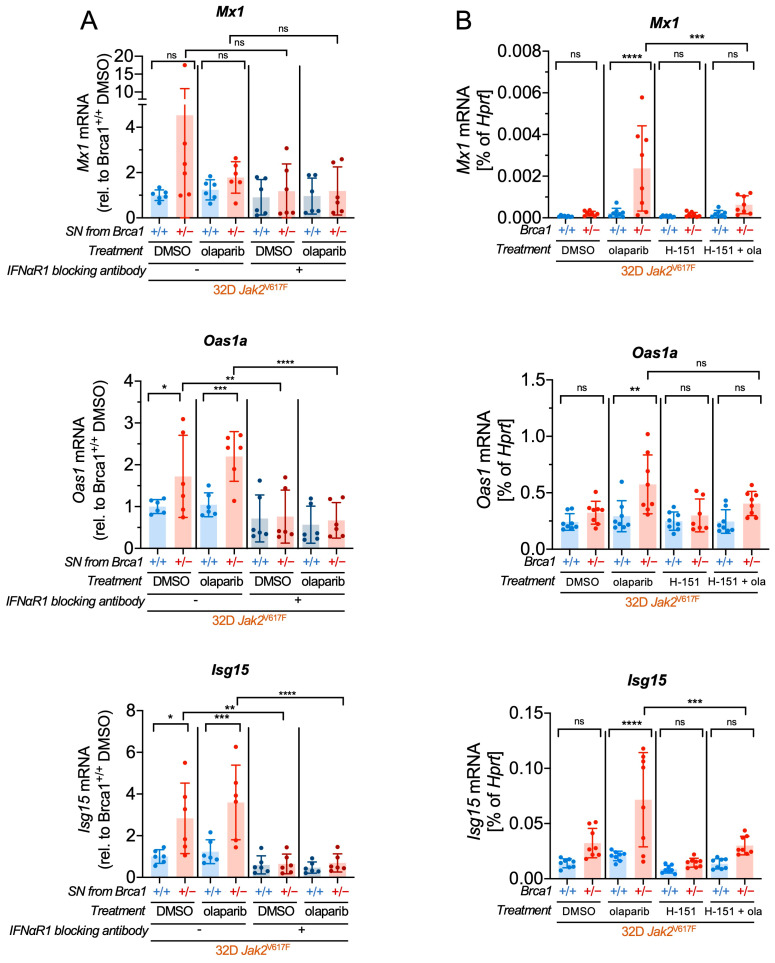
cGAS-STING pathway activation augments the production of IFNα in *Brca1^+/−^ Jak2*V617F cells. (**A**) *Jak2*V617F *Brca1*^+/−^ and *Brca1*^+/+^ cells were first treated for 24 h with olaparib (10 µM) and then cultivated in new medium for 4 h. The supernatant (SN) was added to new *Brca1*^+/+^ cells for 24 h, and transcriptional levels of the interferon-stimulated genes (ISGs) *Mx1*, *Oas1a,* and *Isg15* were analyzed by RT-qPCR. As a negative control, the cells were incubated with an IFNαR1 blocking antibody (*n* = 3 with two clones each). (**B**) *Jak2*V617F *Brca1*^+/−^ and *Brca1*^+/+^ cells were treated with olaparib (ola, 10 µM) and the STING inhibitor H-151 (0.75 µM) for 24 h, and gene expression of *Mx1*, *Oas1a* and *Isg15* was analyzed by RT-qPCR (*n* = 4 with two clones each). Statistical analysis was performed using one-way ANOVA. Data are presented as mean ± SD and significances defined as: * *p* < 0.05, ** *p* < 0.01, *** *p* < 0.001, and **** *p* < 0.0001, ns means no significance.

## Data Availability

The datasets used and/or analyzed during the current study are available from the corresponding author on reasonable request.

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
