# Peer review of "Exploiting Synthetic Lethality between Germline BRCA1 Haploinsufficiency and PARP Inhibition in JAK2V617F-Positive Myeloproliferative Neoplasms"

_ijms, 2023, doi:10.3390/ijms242417560_

Round 1

Reviewer 1 Report

Comments and Suggestions for Authors

PARP inhibitors induce synthetic lethality in BRCA1/2-deficient cancer cells while maintaining low toxicity to normal tissue cells. Over the past two decades, PARP inhibitors have received FDA approval for the treatment of breast cancer, ovarian cancer, and pancreatic cancer in patients with BRCA1 or BRCA2 deficiencies. The promising therapeutic effects of PARP inhibitors have spurred their exploration for potential application in other cancer types.

In the manuscript titled 'Exploiting Synthetic Lethality between Germline BRCA1 Haploinsufficiency and PARP Inhibition in JAK2V617F-Positive Myeloproliferative Neoplasms,' Max Bermes et al. discovered that the PARP inhibitor olaparib causes synthetic lethality in JAK2V617F cells with additional haploinsufficiency of BRCA1. Moreover, they observed that olaparib treatment induced more double-strand DNA breaks and activated the cGAS-STING pathway in Brca1+/- Jak2V617F cells, leading to an increased secretion of IFNα. They proposed that the combination of olaparib and IFNα treatment could serve as a potential synergistic therapeutic strategy for MPN patients with mutations in double-strand repair genes.

Major comments:

1.    Please quantify the proportion of γH2AX foci without a corresponding RAD51 focus in Figure 2A.

2.    Lines 162-176: While the MTT assay measures cell metabolic activity, it actually reflects the cell viability within a well under different treatments. Using "metabolic activity" to interpret the MTT results might lead to confusion among readers regarding the metabolic activity of each cell.

3.    Figure 3 and 4 measure the viability with different methods. Consider combining them and moving some of the figures to supplementary figures.

4.    Please conduct the same MTT assay with increasing concentrations of olaparib in Jak2V617F cells as you did for IFNα in Figure 4A.

Minor comments:

1.    Please include representative images of γH2AX FACS for Figure 1A.

2.    Line 123: The reference figures should be Fig. 1B and D.

3.    Lines 158-159: The figure legends are contradictory to the figures.

4.    Figure 7A: Labeling the supernatant source with only gene names and genotypes might lead to confusion with the cell lines.

5.    Typo IFNa >> IFNα

Comments on the Quality of English Language

Minor editing of English language required

Author Response

Major comments:

  1. Please quantify the proportion of γH2AX foci without a corresponding RAD51 focus in Figure 2A.

We thank the reviewer for this suggestion. We have now included a quantification of the number of γH2AX foci that did not co-localize with a corresponding RAD51 focus. This analysis is now included as Fig. S1K in the supplemental material and mentioned in the result section.

  1. Lines 162-176: While the MTT assay measures cell metabolic activity, it actually reflects the cell viability within a well under different treatments. Using "metabolic activity" to interpret the MTT results might lead to confusion among readers regarding the metabolic activity of each cell.

We have rephrased the sentence, omitting the term “metabolic activity”.

  1. Figure 3 and 4 measure the viability with different methods. Consider combining them and moving some of the figures to supplementary figures.

In response to the reviewer's recommendation, we have combined Figures 3+4 into a single Figure (now labeled as Figure 3). Concurrently, the original content of Figure 3B and Figure 4A+B has been relocated to the supplemental material (Figure S2).

  1. Please conduct the same MTT assay with increasing concentrations of olaparib in Jak2V617F cells as you did for IFNα in Figure 4A.

We have now performed this experiment with two clones of each genotype in three independent experiments. The results are now supplied in the Supplementary Figure S2D. The data aligns well with our previous findings and demonstrates the significantly elevated responsiveness of Brca1+/- cells to different concentrations of olaparib between 5 and 20 µM (5 µM: p<0.05; 10 µM: p<0.0001; 20 µM: p<0.0001).

Minor comments:

  1. Please include representative images of γH2AX FACS for Figure 1A.

We have added a representative FACS plot, now available in Supplementary Figure 1.

  1. Line 123: The reference figures should be Fig. 1B and D.

We have updated the figure reference as suggested.

  1. Lines 158-159: The figure legends are contradictory to the figures.

We apologize for the oversight and thank the reviewer for bringing it to our attention. The error has now been corrected, while we have re-structured the figures, as detailed in major point 3.

  1. Figure 7A: Labeling the supernatant source with only gene names and genotypes might lead to confusion with the cell lines.

We agree with the reviewers and have now included an abbreviation for supernatant (“SN”) in front of the genotypes in Figure 7A (now Figure 6A), and the explanation of the abbreviation is now included in the figure legend.

  1. Typo IFNa >> IFNα

We have now adapted the spelling of the IFN-alpha abbreviation throughout the manuscript and the supplemental material for improved consistency.

Reviewer 2 Report

Comments and Suggestions for Authors

Max Bermes et al. are reporting a comprehensive chemical biology study to suggest that the combination of olaparib and IFNa presents a promising therapeutic strategy for MPN patients. The study is generally well-developed and carefully carried out. Conclusions are supported with data and evidences. The reviewer considers this manuscript to be well-written. The following are some questions to share.

1. Beyond the 32D cell line, have authors considered or tested this therapeutic strategy in other preclinical models, such as animal models or primary patient-derived cells? Or in another word, how well do the 32D Jak2V617F cell line findings translate to human MPN? Are there any known differences between the cell line and the human disease that could impact the validity of the results?

2. The frequency of germline mutations in DSB repair genes in the general population is noted, but is there evidence to suggest this frequency is consistent within the MPN patient population, given that MPN is a rare disorder?

And a minor question on MTT assay specificity. 

3. In the MTT assays comparing Jak2V617F to Jak2WT cells, was there a control for non-specific cytotoxic effects of olaparib and IFNa? What controls are in place to ensure the effects are specific to the synthetic lethality mechanism rather than off-target effects?

Overall, a minor revision would be suggested.

Comments on the Quality of English Language

N/A

Author Response

We appreciate the positive evaluation of our manuscript by the reviewer. We have thoroughly studied and addressed the questions raised and incorporated the suggested revisions into the manuscript, as outlined below.

  1. Beyond the 32D cell line, have authors considered or tested this therapeutic strategy in other preclinical models, such as animal models or primary patient-derived cells? Or in another word, how well do the 32D Jak2V617F cell line findings translate to human MPN? Are there any known differences between the cell line and the human disease that could impact the validity of the results?

Considering the rarity of MPNs and the low prevalence of BRCA1 mutations in the general population, the identification of MPN patients harboring BRCA1 mutations is challenging. We have initiated an extensive screening initiative focused on familial MPN patients, focused on patients with familial MPNs, aiming to identify individuals with the rare combination of both MPN and BRCA1 mutations. Subsequently, we plan to conduct validating experiments using blood cells derived from these rare cases. We are also developing a Brca1 haploinsufficient mouse model. This model is designed to provide in vivo data that complements our mechanistic study, offering insights into the concept of synthetic lethality between BRCA1 haploinsufficiency and PARP inhibition in JAK2V617F-positive cells within a whole organism context.

To address the translational relevance of our findings, we have incorporated a more comprehensive discussion in the manuscript, acknowledging the challenges associated with the scarcity of suitable human samples and outlining our approach to bridge the gap between the 32D cell line findings and the potential clinical application of our therapeutic strategy.

  1. The frequency of germline mutations in DSB repair genes in the general population is noted, but is there evidence to suggest this frequency is consistent within the MPN patient population, given that MPN is a rare disorder?

The prevalence of BRCA1 germline mutations in the general population is estimated to be around 0.2-0.3%. In a whole-exome sequencing study with familial MPN patients, we detected BRCA1 mutations in two out of ten individuals of five families. While acknowledging the limitation of the small sample size, these findings suggest a potential over-representation of BRCA1 mutations in familial MPN cases, hinting at the possibility that genomic instability may contribute to the acquisition of driver mutations. Nevertheless, the significance of these observations requires further investigation with a larger sample size, a task currently underway in our group.

  1. In the MTT assays comparing Jak2V617F to Jak2WT cells, was there a control for non-specific cytotoxic effects of olaparib and IFNa? What controls are in place to ensure the effects are specific to the synthetic lethality mechanism rather than off-target effects?

We have hypothesized that the concept of synthetic lethality between PARP inhibition and Brca1 haploinsufficiency is especially exploitable in Jak2V617F-positive cells, given their heightened DNA damage. To address this, MTT assays were conducted not only with Jak2V617F cells but also with Jak2WT cells to compare treatment efficacy. We demonstrate that Jak2WT Brca1+/- cells were less susceptible to olaparib (or IFNα, which is known to induce DNA stress) than Jak2V617F Brca1+/- cells, implying that the observed effects are specific to Jak2V617F. However, acknowledging that Jak2WT cells were also affected to some extent by the treatment, non-specific cytotoxic effects cannot be entirely excluded. Further studies should evaluate the dosing regime to mitigate potential off-target effects. The primary focus of our study was to elucidate mechanistic implications rather than address safety concerns.

Reviewer 3 Report

Comments and Suggestions for Authors

Major Concerns

1.     In figure 1. C), the usage of DMSO only was shown to significantly augment the γH2AX expression in BRCA1+/- condition. The Olaparib-treated group should have the spillover effect of DMSO usage as Olaparib is soluble in DMSO and used as such condition. Did authors consider performing a synergy estimation of DMSO and Olaparib in Olaparib treated group and subtracting the DMSO effect?

2.     γH2AX foci formation is the effective DNA damage readout rather than its expression. Through figure 2 and the statement in line 132, it was shown that BRCA1+/- cells did not showed significantly elevated γH2AX foci counts/cell. The numerical induction may still represent the additive effect of DMSO usage as a solvent of Olaparib. How can authors explain this phenomenon? Also, based on these results, the claims should be modest and should be reflected in the Abstract and Conclusion sections.

3.     The mRNA expression of p16 and p21 pathway genes may not be the best readout of cell cycle analysis. I would suggest the authors to provide Flow cytometry-based cell cycle phage distribution among the treatment/genotype groups. The mRNA expression data could be supportive data with the Flow cytometry data.

Minor concerns

1.     The details of the antibodies used in Western blot experiments are not mentioned in the ‘Materials and Methods’ section. I recommend that authors to tabulate all the antibodies and reagents used in this manuscript with the name of the reagents, supplier, catalog no. RRID no, dilution used for that particular experiment, any special treatment (e.g. antigen retrieval condition, etc.).Please include these in Supplementary Table 1.2.

2. Why authors used the 32D cell line of murine origin instead of the human cell lines (e.g, HEL, UKE-1) as the preliminary data of the study could be translatable to the potential clinical application?

Comments on the Quality of English Language

The manuscript should be reviewed for common grammatical and phrasing errors throughout.

Author Response

Major Concerns

  1. In figure 1. C), the usage of DMSO only was shown to significantly augment the γH2AX expression in BRCA1+/- condition. The Olaparib-treated group should have the spillover effect of DMSO usage as Olaparib is soluble in DMSO and used as such condition. Did authors consider performing a synergy estimation of DMSO and Olaparib in Olaparib treated group and subtracting the DMSO effect?

The designation of +/- on the x-axis in Fig. 1C does not represent the addition of the corresponding treatment, but the Brca1 genotype, as indicated by the axis legend on the left. Consequently, the data show that γH2AX phosphorylation is elevated in Brca1+/- cells compared to Brca1+/+ cells. As shown in Fig. 1A, 0.01% DMSO (the mass percentage that we have used) alone does not induce DNA double strand breaks, when compared to the ATM inhibitor (KU55933, negative control). This finding aligns with existing literature, which indicates that substantially higher concentrations are required to induce DNA damage.

  1. γH2AX foci formation is the effective DNA damage readout rather than its expression. Through figure 2 and the statement in line 132, it was shown that BRCA1+/- cells did not showed significantly elevated γH2AX foci counts/cell. The numerical induction may still represent the additive effect of DMSO usage as a solvent of Olaparib. How can authors explain this phenomenon? Also, based on these results, the claims should be modest and should be reflected in the Abstract and Conclusion sections.

We acknowledge that, although γH2AX levels were significantly increased in Western blot and flow cytometry readouts, no significant changes were observed by confocal microscopy, albeit a trend was apparent. This might be attributed to the lower sensitivity of the confocal microscopy assay, given the necessity for the phosphorylation of a multitude of H2A proteins to form a focus detectable through this method. However, we have now utilized a similar approach as for RAD51 quantification and determined the percentage of cells with ≥10 γH2AX foci revealing an increase for Brca1+/- cells (p=0.0712; new Fig. 2B). To enhance the transparency of our findings, we have now included the exact p-values derived from the three different methods in the abstract and have made corresponding adjustments to the discussion section for clarity.

  1. The mRNA expression of p16 and p21 pathway genes may not be the best readout of cell cycle analysis. I would suggest the authors to provide Flow cytometry-based cell cycle phage distribution among the treatment/genotype groups. The mRNA expression data could be supportive data with the Flow cytometry data.

We appreciate the suggestion and fully agree that an additional flow cytometry-based assay will greatly strengthen our assertion that the combined treatment with olaparib and IFNα induced a cell cycle arrest selectively in Brca1 haploinsufficient cells. To enhance the robustness of our findings, we have incorporated a flow cytometry-based cell cycle analysis into Fig. S5 that strongly corroborates our findings from the RT-qPCR experiments.

Minor concerns

  1. The details of the antibodies used in Western blot experiments are not mentioned in the ‘Materials and Methods’ section. I recommend that authors to tabulate all the antibodies and reagents used in this manuscript with the name of the reagents, supplier, catalog no. RRID no, dilution used for that particular experiment, any special treatment (e.g. antigen retrieval condition, etc.).Please include these in Supplementary Table 1.2.

In response to the reviewer's suggestion, we have made several enhancements. Specifically, we have added a statement in the “Materials and Method” section directing readers to the antibody list in the supplements for comprehensive details. To enhance alignment with the manuscript references, we have reorganized the supplementary tables 1.1 and 1.2. We have introduced an additional column providing the catalogue numbers or RRID for better identification. Furthermore, to provide more transparency, information regarding the dilutions employed for the respective techniques has been incorporated.

  1. Why authors used the 32D cell line of murine origin instead of the human cell lines (e.g, HEL, UKE-1) as the preliminary data of the study could be translatable to the potential clinical application?

The rationale behind our decision to employ 32D cells is that we wished to mechanistically explore the synthetic lethality between PARP inhibition and Brca1 haploinsufficient cells, considering the added complexity of the oncogenic JAK2V617F mutation as an additional stressor. 32D cells transduced with Jak2V617F provide the advantage of Jak2WT-transduced cells serving as an ideal isogenic control. Our choice to work with this model is particularly important for investigating synthetic lethality enabling an estimation of the impact of the different treatment conditions on non-mutated cells. In contrast, human cell lines such as HEL, SET2, or UKE-1 lack this isogenic control feature, limiting their suitability for such assessments. To elucidate the clinical translatability of our findings, we plan to conduct future experiments using animal models, as well as MPN patient-derived peripheral blood cells and progenitor cells differentiated from induced pluripotent stem cells harboring JAK2V617F mutations and concurrent BRCA1 haploinsufficiency.

Round 2

Reviewer 3 Report

Comments and Suggestions for Authors

Thank you for addressing the concerns properly.